# Convergent Mutations and Single Nucleotide Variants in Mitochondrial Genomes of Modern Humans and Neanderthals

**DOI:** 10.3390/ijms25073785

**Published:** 2024-03-28

**Authors:** Renata C. Ferreira, Camila R. Rodrigues, James R. Broach, Marcelo R. S. Briones

**Affiliations:** 1Center for Medical Bioinformatics, Federal University of São Paulo, São Paulo 04039032, SP, Brazil; recarmona@gmail.com; 2Graduate Program in Microbiology and Immunology, Federal University of São Paulo, São Paulo 04039032, SP, Brazil; camila_rodrigues92@hotmail.com; 3Department of Biochemistry, Institute for Personalized Medicine, Pennsylvania State University College of Medicine, 500 University Drive, Hershey, PA 17033, USA; jbroach@pennstatehealth.psu.edu

**Keywords:** human genome, mitochondrial genome, neanderthal admixture, SNPs, ancient DNA

## Abstract

The genetic contributions of Neanderthals to the modern human genome have been evidenced by the comparison of present-day human genomes with paleogenomes. Neanderthal signatures in extant human genomes are attributed to intercrosses between Neanderthals and archaic anatomically modern humans (AMHs). Although Neanderthal signatures are well documented in the nuclear genome, it has been proposed that there is no contribution of Neanderthal mitochondrial DNA to contemporary human genomes. Here we show that modern human mitochondrial genomes contain 66 potential Neanderthal signatures, or Neanderthal single nucleotide variants (N-SNVs), of which 36 lie in coding regions and 7 result in nonsynonymous changes. Seven N-SNVs are associated with traits such as cycling vomiting syndrome, Alzheimer’s disease and Parkinson’s disease, and two N-SNVs are associated with intelligence quotient. Based on recombination tests, principal component analysis (PCA) and the complete absence of these N-SNVs in 41 archaic AMH mitogenomes, we conclude that convergent evolution, and not recombination, explains the presence of N-SNVs in present-day human mitogenomes.

## 1. Introduction

Comparative analyses of present-day human genomes and paleogenomes led to the proposal that *Homo sapiens neanderthalensis* (Neanderthal) contributed to the makeup of the *Homo sapiens sapiens* (present-day human) genome [1,2,3,4]. The current interpretation of genomic signatures is that Neanderthal contributions are different in European, East Asian and African lines of descent, with a higher frequency of Neanderthal segments in Asians and Europeans and lower frequencies in Africans [1]. Paleolithic *Homo sapiens sapiens*, or archaic anatomically modern humans (AMHs), migrated from Africa into the Middle East and Europe in the last 50,000 years and intercrossed with Neanderthals, which explains the presence of Neanderthal signatures in extant human genomes [3,4]. The spatio-temporal overlap of Neanderthals and archaic AMHs is estimated to have been approximately 22,000 years, since the first archaic AMHs arrived in Europe around 50,000 years ago and the last Neanderthal remains (in Spain) date back 28,000 years [5,6].

Although there is evidence for Neanderthal contributions to the present-day human nuclear genome, it has been proposed that there is no Neanderthal contribution to present-day human mtDNA, or the mitochondrial genome (mitogenome) [7]. It is widely accepted that the human mitogenome is exclusively inherited from the mother and lacks recombination [8]. Because of the matrilineal inheritance of the mitogenome, this implies that the intercrosses occurred exclusively between Neanderthal males and archaic AMH females or that crosses between archaic AMH males and Neanderthal females were extremely rare. Another possibility is that crosses between archaic AMH males and Neanderthal females produced such unfavorable trait combinations, due to mitonuclear incompatibility [9], that none of their descendants left marks in present-day human populations.

The lack of recombination in the human mitogenome is still a matter that is hotly debated, with published evidence both supporting it and rejecting it [10,11]. The main problem posed by assuming a complete lack of human mitogenome recombination is Muller’s ratchet, the theoretical concept in population genetics that predicts the accumulation of harmful mutations in a population over time [12,13]. Muller’s ratchet highlights the importance of recombination in the long-term maintenance of genetic diversity and the prevention of genetic deterioration in populations. Recombination allows for the removal of harmful mutations and shuffling of genetic material, thereby alleviating the effect of Muller’s ratchet [14].

If Neanderthals left traces of their mitogenome in the present-day human mitogenome due to intercrosses, as observed for the nuclear genome, we would expect that polymorphisms, or Neanderthal-specific signatures, would be observed in present-day human mitogenomes but not in archaic AMHs. To test this hypothesis, we analyzed the mitogenomes of present-day humans, archaic AMHs and Neanderthals and found 66 single nucleotide variants (SNVs), or signatures, in present-day human mitogenomes that were also present in Neanderthals but not in archaic AMHs. The presence of these 66 SNVs (N-SNVs) can be explained either by rare mitogenome recombination or convergent mutations. The archaic AMH samples selected for analysis are approximately the same age as the Neanderthal mitogenomes, and therefore represent, in theory, an archaic AMH ensemble that could have overlapped with Neanderthals in Europe, the Middle East and Central Asia. This also allows for an assessment of homoplasy because the Neanderthal and archaic AMH sequences here considered are of approximately the same age. We investigated the distribution of these SNVs in different human haplogroups and performed principal component analysis (PCA) and recombination tests to evaluate whether recombination or convergent mutations (homoplasy) might explain the N-SNVs in present human mitogenomes.

## 2. Results

### 2.1. Distribution of Neanderthal SNVs

We aligned 102 mitogenomes comprising 9 Neanderthal samples (Appendix A), 41 archaic AMH samples (Appendix A) and 52 present-day humans with representatives of the major worldwide mitochondrial haplogroups (Appendix A). This alignment contained the revised Cambridge Reference Sequence (rCRS) used as a numbering reference for all polymorphisms identified [15]. From this alignment, 918 polymorphic positions (SNVs) were identified. Among these, 66 SNVs occurred in all present-day humans examined and in at least one Neanderthal SNV (N-SNV) (Figure 1, Table 1 and Appendix A). In 13 positions, a subset of present-day humans carried a variant identical to at least one Neanderthal sequence position and at least one sequence of an archaic AMH. Neanderthals differed from present-day humans and archaic AMHs in 175 positions. In 11 positions, the archaic AMHs differed from all other sequences. Present-day humans have 653 exclusive SNVs, which correspond to inter-haplogroup sequence differences.

To depict the distribution of Neanderthal SNVs, or N-SNVs, in different human mitochondrial haplogroups, we constructed a heat map of N-SNVs (Figure 2). Most N-SNVs are concentrated in the D-loop, followed by 12SrDNA and 16SrDNA. Among tRNAs, N-SNVs were found only in isoleucine, asparagine and cysteine tRNAs. In *COX2,* only one N-SNV was found in the haplogroup R0a.

Of the 66 N-SNVs identified, 20 are common to modern African and Eurasian haplogroups, 25 are exclusive to African haplogroups and 21 are exclusive to Eurasian haplogroups. In Figure 3, the distribution of N-SNVs is depicted in the human mitogenome map. The distribution reveals that 11 Eurasian N-SNVs are in coding regions, 3 in rRNA genes and 1 in a tRNA gene.

The distribution of N-SNVs in modern haplogroups and archaic AMHs can be summarized in five patterns (columns) among the mitogenomes analyzed here (Figure 4). Pattern 1 consists of 13 N-SNVs that are present in all five clades. Pattern 3 consists of 20 N-SNVs that are present in Neanderthals and in present-day humans but not in archaic AMHs. Pattern 4 consists of 25 N-SNVs that are present in Neanderthals and only in the present-day human haplogroup L, while Pattern 5 consists of 21 N-SNVs that are present in Neanderthals and only in present-day Eurasian haplogroups. The 66 N-SNVs analyzed here are in Patterns 3, 4 and 5, which excludes their presence in archaic AMHs and thus they are likely signals of introgression of convergent mutations. More importantly, no SNVs are identical between Neanderthals and archaic AMHs but absent from the present-day mitogenomes of either African or Eurasian haplogroups (Figure 4, Pattern 2). This further suggests that the presence of these N-SNVs reflects either horizontal transfer or a significant number of reverse/convergent substitutions.

### 2.2. Disease-Associated N-SNVs

Among the 66 N-SNVs, 7 are associated with diseases as depicted in Table 2. Of note, 4 of these disease-associated N-SNVs were observed in African haplogroups (L0, L1, L2, L3, L4, L5 and L6) and 3 were observed exclusively in Eurasian haplogroups. The most common diseases associated with N-SNVs are neurological disorders and tumors. One N-SNV, in Position 15,043 and associated with depression, was also found in one archaic AMH. Although not considered a bona fide N-SNV based on our exclusion criteria, it is relevant because chronic depression has been associated with Neanderthal introgression in modern humans [16]. An N-SNV in *ND2* (position 5460) causes an amino acid change from alanine to threonine. This SNV was initially associated with Alzheimer’s disease and Parkinson’s disease, although subsequent studies have not replicated these associations [17,18,19]. Other disease-associated N-SNVs are in the D-loop, 16S rRNA and tRNA-Cys (Table 2). The prevalence of diseases associated with N-SNVs in Table 2 are as follows (data from https://vizhub.healthdata.org/gbd-compare/, accessed on 31 August 2019): bipolar disorder = 596 cases in 100,000 persons (596/100,000), Parkinson’s’ disease = 111/100,000, Alzheimer’s disease = 588/100,000, melanoma = 30.42/100,000, ovarian cancer = 17.71/100,000, lung cancer = 43/100,000, prostate cancer = 129/100,000 and stomach cancer = 36.9/100,000. Also associated are cycling vomiting syndrome 3.2/100,000 [20], deafness = 110/1000 [21] and glioblastoma = 10/100,000 [22].

### 2.3. Haplogroups of Paleogenomes

The mitogenomes of Neanderthals and archaic AMHs were classified in haplogroups according to sequence similarity with extant human mitogenomes using Haplogrep 2 [20] (Table 3). In total, 83% of archaic AMH mitogenomes belong to the haplogroup U (45% haplogroup U5 and 16% to haplogroup U2) which is consistent with U being the oldest European haplogroup. Among the nine Neanderthal mitogenomes, the seven more recent genomes can be classified as the haplogroup H1 (European) while the two oldest can be classified as the haplogroup L (African) (Table 3). N-SNVs at Positions 16,278 and 16,298 are associated with the intelligence quotient [24]. N-SNV 16,278 is found in African haplogroups (L0, L1, L2, L5 and L6) and two Eurasian haplogroups (X3, U2c and P2) and in all Neanderthal sequences, while N-SNV 16,298 is found only in the Eurasian–Native American haplogroups (V1, V2, M8, C1, C4, C7 and Z1) and only in the Altai Neanderthal.

### 2.4. PCAs of Neanderthal and Human Mitogenomes

Principal component analysis (PCA) of the whole mitochondrial genome shows four clusters: (1) the modern haplogroups including ancient *H. sapiens* (archaic AMHs), (2) the L haplogroup cluster, (3) the Neanderthal Altai-Mezmaskaya L-like cluster and (4) the Neanderthal H-like group (Figure 5). The PCA of the segment corresponding to the ribosomal RNA gene proximal half produces a pattern that approximates the present-day L haplogroup to the Neanderthal Altai-Mezmaskaya L-like group, which suggests a possible introgression point. PCA of the ribosomal RNA gene distal half suggests an opposite pattern, with the haplogroup L being closer to Neanderthal, although not as close as shown in Figure 5B. This analysis justifies the following testing for recombination in an attempt to estimate whether reverse/convergence mutations could be confounded with recombination as discussed above.

### 2.5. Bootscan Analysis of Mitogenomes

We tested potential recombination in our dataset with bootscan. We used a different set of parental sequences depending on the query mitogenome (Figure 6). Although the bootscan analysis indicates that there are potential small recombination points, no extensive blocks of recombinant molecules were detected as is typical with bona fide recombinants. Upon deeper analysis, we observed that bootscan considers the Neanderthal-specific signatures, such as in L haplogroups, as recombination points. Although the bootscan putative recombination segments are above the bootstrap threshold, we do not consider this as solid evidence for recombination because the segments between the Neanderthal signatures are almost identical. Bootscan analysis excluded human–Neanderthal recombination in the rCRS sequence (Figure 7). The sensitivity of bootscan to substitution models and alignment methods was assessed by comparing the same set query-parentals with different parameters (Figure 8), revealing minor profile alterations. The alignment parameters are not so critical in this case because the sequences are extremely conserved (918 polymorphic positions in 16,569 bp, or 5.54% divergence). Although indels are present in the alignments, 99% are located near the H promoter in the D-loop region. These are automatically excluded in phylogeny inference algorithms and therefore have no weight in the bootscan results. The “positional homology” is therefore solid, particularly in coding domains and regions without repeats in non-coding domains. The Neanderthal signatures are in unambiguously aligned segments.

## 3. Discussion

In the present study, we observe Neanderthal signatures in modern human mitochondrial genomes. These Neanderthal signatures (N-SNVs) are variants present in all Neanderthal mitogenomes and in present-day human mitogenomes but not in archaic AMHs. The recombination tests presented here (bootscan analysis) show that recombination does not explain the presence of these 66 SNVs, or N-SNVs, in all modern mitochondrial haplogroups. The topic of recombination in the human mitogenome is controversial and evidence supporting it as well as evidence refuting it have been reported [10,11]. We analyzed an alignment of 102 mitogenomes, which contained 918 polymorphic positions, by bootscan tests and PCA with the idea to evaluate whether the presence of N-SNVs could be explained solely by homoplasy or if recombination would be necessary. Although a high homoplasy rate could explain the N-SNVs in the D-loop, we asked if recombination could be associated with N-SNVs in coding regions, especially the nonsynonymous changes in Positions 5460 (*ND2*, rs3021088), 7146 (*COX1*, rs372136420), 7650 (*COX2*, without rs, ACC>ATC), 9053 (*ATP6*, rs199646902), 13,105 (*ND5*, rs2853501), 13,276 (*ND5*, rs2853502) and 14,178 (*ND6*, rs28357671). Our bootscan analyses suggested that all these N-SNVs could be explained by homoplasy in the last 40,000 years and that these changes occurred only in the present-day lineages and not in any of the archaic AMH mitogenomes analyzed.

The mitogenomes analyzed were from present-day humans, archaic AMHs and Neanderthals as detailed above. Based on populational data, Sykes (2001) estimated that a single observed change in comparative mitogenomics corresponds to 10,000 years of divergence. More modern estimates of the mitogenome clock, based on ancient DNA data, range from 2.14 × 10^−8^ to 2.74 × 10^−8^ substitutions per site per year [26,29,33,34], which gives approximately 4.14 substitutions in mitogenome in 10,000 years. Therefore, according to populational estimates, the 918 polymorphisms would have occurred in a period of 9.18 million years and the 66 N-SNVs in 660,000 years. If ancient DNA estimates are considered, the 918 polymorphisms would correspond to 2.21 million years of evolution and the 66 N-SNVs would have occurred in the last 159,420 years. Therefore, all 66 N-SNVs are a product of random changes or simple homoplasy not observed in any of the 41 samples of archaic AMHs. This suggests that homoplasy, parallel with Neanderthals, occurred only in the modern lineage, and not in archaic AMHs. Tests for recombination using bootscan (Figure 6) indicated that in 11 positions the bootstrap supports recombination but does not explain all N-SNVs. It is important to notice that Posada and Crandall [35] explicitly tested how homoplasy could confound recombination tests and concluded that in extreme levels of rate variation (α = 0.05), recombination tests would produce false positives, which fits with the mitogenome mutational load and among-site rate variation. 

The analysis of Neanderthal mitochondrial genomes presented here revealed four derived amino acid changes that modern humans carry in the *COX2* gene as compared to Neanderthals and other ape outgroups [30]. However, the same four amino acid changes can also be found in macaques, which suggests that there is no need to invoke mitochondrial recombination. The divergence between the *H. sapiens* lineage and macaques lineage is 30.5 (26.9–36.4) million years [36], while the divergence we are dealing here is between 300,000 and 28,000 years. As is known, homoplasy significantly increases with long divergence times which produces the effect of long branch attraction in phylogenies [37]. For example, one of the mutations would be the macaques’ parallel mutation in the *COX2* gene m.7650C>T. This mutation is found in Neanderthals (except Mezmaskaya), Denisovans, the modern R0 haplogroup, gorilla, chimpanzee, and bonobo, but it is not found in any archaic AMH samples. This pattern more likely suggests that it is highly conserved, inherited by early hominins from apes and secondarily lost in one Neanderthal lineage and in almost all *Homo sapiens* lineages, except for the R0 haplogroup. The reappearance of this mutation in the very old R0 haplogroup has two possible explanations: (1) a back mutation reverting to the same ancestral state instead of changing to any of the other three possible bases or (2) a mutation acquired by recombination via a Neanderthal female with introgression in one of the *Homo sapiens* lineages. In our study, we present a bootstrap recombination analysis (bootscan) that shows a recombination point with bootstrap support in the region encompassing m.7650C>T (Figure 6A,E,F). Based on this test alone, it seems that at least regarding this mutation a very rare recombination event could have happened, although a back mutation cannot be completely excluded. Another possible argument is that the Neanderthal signatures are in fact character states conserved since the last common ancestor of Neanderthals and present-day *Homo sapiens* (e.g., *Homo erectus*), but this would not be consistent with the absence of these signatures in ancient *H. sapiens* mitochondrial genomes (Figure 1). 

With regard to the presence of N-SNVs in African haplogroups, we note that a back-to-Africa hypothesis has been proposed in which humans from Eurasia returned to Africa and impacted a wide range of sub-Saharan populations [38]. Our data suggest that Neanderthal signatures might be present in all major African haplogroups, which is consistent with the “Back to Africa” contribution to the modern mitochondrial African pool. The preponderance of N-SNVs in the D-loop is observed mostly in African haplogroups. In Eurasian haplogroups, we observe important changes in coding regions as demonstrated in Figure 2 and Figure 3. Although our data suggest that convergent mutations explain the N-SNVs here observed, mitochondrial recombination is not so rare as to completely exclude it as a potential mechanism in human mtDNA mutational patterns [39,40]. 

Our observations suggest that crosses between archaic AMH males and Neanderthal females left significantly fewer descendants than the reverse crosses (Neanderthal males and AMH females). Although it has been generally accepted that recombination does not occur in the human mitochondrial genome, evidence of mitochondrial recombination has been reported [10,41]. A scenario with a complete absence of recombination presents a problem to explain how the human mitochondrial genome would escape Muller’s ratchet and therefore avoid its predicted “genetic meltdown” [14]. However, even minimal recombination is sufficient to allow escape from Muller’s ratchet [42], and this could be the case regarding the human mitochondrial genome. For example, recombination has been simulated along a chromosome of 1000 *loci* to estimate the amounts of recombination required to halt Muller’s ratchet and the drift-catalyzed fixation of deleterious mutations. For a population size of *N* < 100, a recombination rate equivalent to one crossover per chromosome per 100 generations (10^−5^/locus/generation) countered Muller’s ratchet effectively. This is much lower than the minimum of one crossover per chromosome arm per generation that is assumed to occur in sexual taxa. A higher recombination rate of 10^−4^ can impede the selective interference that would otherwise enhance the fixation of deleterious mutations due to genetic drift [43].

Our data are compatible with a scenario in which archaic AMH–Neanderthal crosses occurred in Europeans, East Asians, and African lines of descent, which is consistent with recent findings in nuclear genomes [44]. However, in the African haplogroups, the crosses between archaic AMH males and Neanderthal females would have a higher frequency than in European lines of descent, where the reverse crosses would be predominant. Based on the comparison of Neanderthal signatures in the nuclear and mitochondrial genome haplogroups, we hypothesize that the African lines of descent would have a higher female Neanderthal contribution, whereas European lines of descent would have a higher male Neanderthal contribution. The fact that archaic AMHs and Neanderthals crossed and produced fertile descendants is evidence that they belong to the same species [2]. Some authors propose that this suggests that *Homo sapiens* emerged independently in Africa, Europe, and Asia [45]. The intercrosses of these three *Homo sapiens* subgroups, and other even deeper ancestors such as Denisovans, in their different proportions and specific signatures, would have produced the extant human genomes.

## 4. Material and Methods

### 4.1. Mitochondrial Genome Sequences

Mitochondrial genomes were downloaded from GenBank for 9 Neanderthal samples (Appendix A) and 41 archaic AMH samples (Appendix A), and 52 sequences of present-day human mitogenomes, representing all major mitochondrial haplogroups, as selected from the PhyloTREE database [46] (Appendix A). The archaic AMH Ust-Ishim sequence was assembled using reads downloaded from Study PRJEB6622 at the European Nucleotide Archive (EMBL-EBI) and assembled using the CLC Genomics Workbench 7 program (https://www.qiagenbioinformatics.com) (Qiagen, Hilden, Germany).

### 4.2. Sequence Alignments

To maintain the reference numbering, sequences were aligned to the revised Cambridge Reference Sequence (rRCS; GenBank accession number NC012920) [16], using the map to reference option implemented in Geneious 10 program (http://www.geneious.com/) (Dotmatics, Boston, MA, USA) [47]. Variants were called using the Geneious 10 program (http://www.geneious.com/). A total of 918 polymorphic positions were found. Neanderthals and ancient and modern humans were screened for disease associations on the MitoMap database website (http://www.mitomap.org/MITOMAP, accessed on 24 January 2024) [23].

### 4.3. Phylogenetic Inference

Position-specific similarities between modern haplogroups and Neanderthals were depicted by cladograms for each of the single 66 variant positions present only in Neanderthals and modern humans and excluding archaic AMHs. Cladograms were generated using a parsimony heuristic search implemented in PAUP v4.1a152 with the default parameters [48]. The proximities of mitochondrial haplogroups in ancient *H. sapiens* and Neanderthals were inferred using Haplogrep 2.1.0 [25]. All 66 cladograms, corresponding to each N-SNV, are available upon request.

### 4.4. Recombination Analysis

Potential recombination between Neanderthals and ancient *H. sapiens* sequences was inferred by a phylogenetic based method implemented by manual bootscan in the recombination detection program (RDP) v.4.87 [49]. The parameters for bootscan analysis were as follows: window size = 200; step size = 20; bootstrap replicates = 1000; cutoff percentage = 70; use neighbor joining trees; calculate binomial *p*-value; model option = Kimura 1980 [50]. For each analysis, a single alignment was created which included the modern haplogroup, all nine Neanderthal sequences and all six archaic AMH sequences. When rCRS was used as the query, two sets of possible parental sequences were selected: either the Neanderthals Mezmaiskaya and Altai and the ancient *H. sapiens* Fumane and Ust-Ishim or only the Neanderthals Feldhofer1, Mezmaiskaya and Vindija 33.16. For the haplogroups L0d1a and L3d3b, possible parental sequences were the Neanderthals Feldhofer1, Mezmaiskaya and Vindija 33.16 and the ancient *H. sapiens* Kostenki 14, Fumane, Doni Vestonice 14 and Tianyuan. For the haplogroups M29a and R0a, possible parental sequences were the Neanderthals Mezmaiskaya and Altai and the ancient *H. sapiens* Kostenki 14 and Doni Vestonice 14. For the haplogroup N1b1a3, possible parental sequences were the Neanderthals Feldhofer1 and Vindija 33.16 and the ancient *H. sapiens* Kostenki 14 and Doni Vestonice 14.

### 4.5. Statistical Analysis

For variant calling (SNVs), three different datasets were used: (1) the whole mitogenome from the 102 sequences’ alignment; (2) the 128 to 315bp fragment; and (3) the 6950 to 7660 bp fragment in the same alignment. All fasta alignments were processed using the MSA2VCF software version 1.0 (https://lindenb.github.io/jvarkit/MsaToVcf.html, accessed on 24 January 2024) to generate the VCF files [51]. The options used on msa2vcf were: --haploid --output. To convert the VCF files to Plink format we used the vcftools package [52]. The whole mitogenome alignment with 103 sequences had 785 SNPs (positions containing gaps in at least one sequence were excluded from the analysis). Both the 128–315 and 6950–7660 fragments had 24 SNPs.

Principal component analysis (PCA) was performed using the PLINK software v1.90b4 [53,54]. PCA figure plotting was performed using Genesis PCA and the admixture plot viewer (http://www.bioinf.wits.ac.za/software/genesis/, accessed on 24 January 2024). The first two principal components were chosen for the comparison of Neanderthal versus archaic AMH versus present-day human mitogenomes.

## 5. Conclusions

The analyses presented here suggest that Neanderthal genomic signatures might have been a product of convergent evolution due to homoplasy and that rare mtDNA recombination events cannot explain the presence of these signatures in modern mtDNA haplogroups. Although evidence for recombination in human mtDNA has been published, its weight in phylogenies and mutational patterns remain controversial. Some authors hypothesize that due to the high mutation rate of mtDNA, reverse compensatory mutations can be confounded with recombination. Our data are consistent with a scenario in which mtDNA recombination is not sufficient to support the contribution of Neanderthal mtDNA to modern human genomes, although rare recombination events might occur in human mtDNA to alleviate the effects of Muller’s rachet. As a future perspective, one possibility would be to construct cell lineages containing synthetic mitogenomes (cybrids) with the Neanderthal SNVs characterized here, to measure metabolic parameters and the different compatibility between nuclear genome alleles and different mitogenome types to assess the possible energetic metabolic characteristics of Neanderthals.

## Figures and Tables

**Figure 1 ijms-25-03785-f001:**
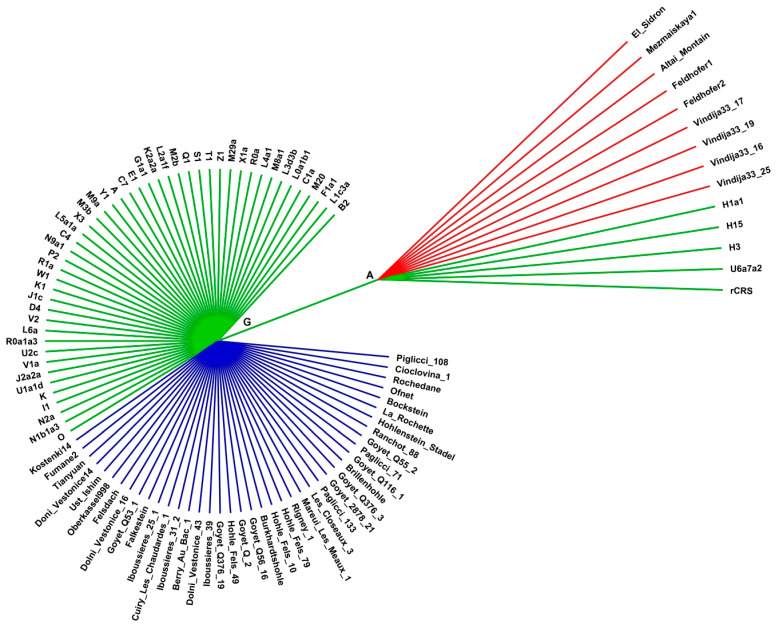
Cladogram of human mitogenome position 2706. This N-SNV is a G-to-A transition in the 16S rRNA gene. This genetic signature (Neanderthal signature 2706G) is present in all Neanderthal sequences (red branches) and in the European haplogroups U (U5a7a2) and H (H1a1, H3, H15), including the revised Cambridge Reference Sequence (rCRS, haplogroup H2a2a1). The Neanderthal signature 2706G is absent in all archaic AMHs from Europe in temporal overlap with Neanderthals (blue branches) and other present-day mitochondrial haplogroups (green branches). The position numbering corresponds to the rCRS positions. This cladogram is representative of the 66 cladograms generated for each N-SNV showing similar topologies.

**Figure 2 ijms-25-03785-f002:**
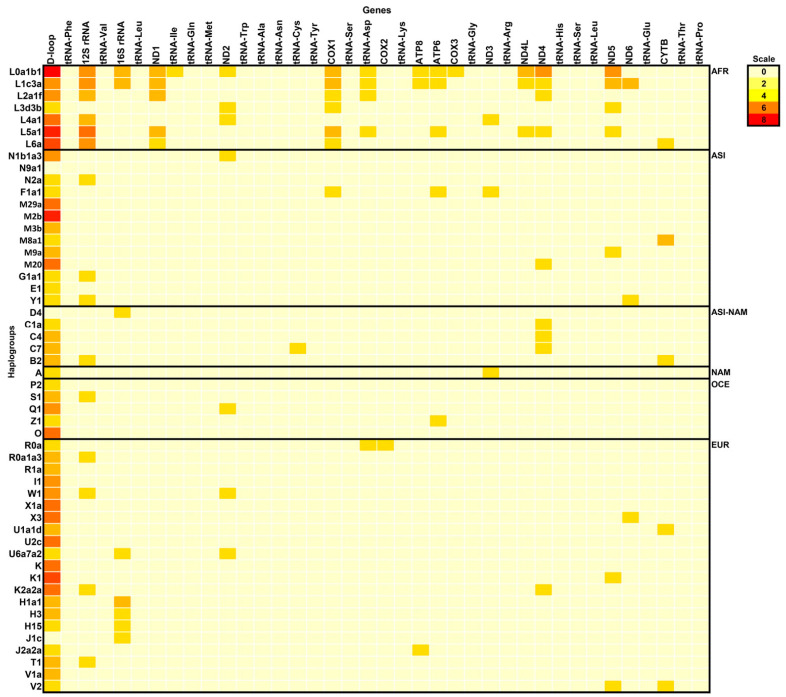
Heatmap of Neanderthal signatures along the mitogenome in different haplogroups. The color scale indicates the number of Neanderthal signatures (N-SNVs) present in modern human mitochondrial haplogroups. These signatures are absent in ancient *H. sapiens* whose time range overlapped with Neanderthals in Europe (from approximately 50,000 to 28,000 years ago, Table 1). AFR = African, ASI = Asian, NAM = Native American, OCE = Oceanian and EUR = European.

**Figure 3 ijms-25-03785-f003:**
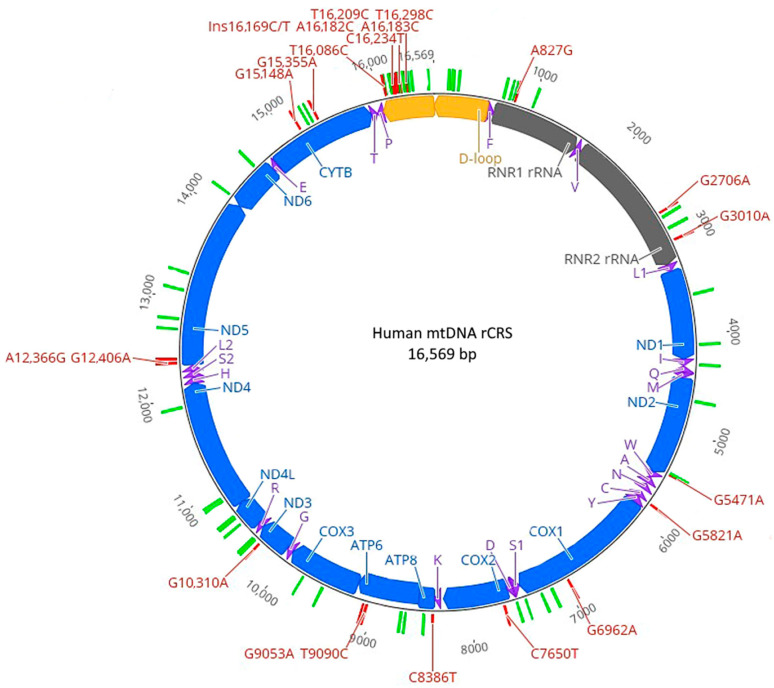
Positions of Neanderthal SNVs (N-SNVs) in the human mitogenome reference map. Blue solid lines indicate protein coding genes, gray solid lines indicate ribosomal RNA genes, purple arrows indicate tRNAs, yellow arrows indicate the D-loop, small green bars indicate Neanderthal SNVs present in African haplogroups, and small red bars indicate N-SNVs exclusive to Eurasian haplogroups.

**Figure 4 ijms-25-03785-f004:**
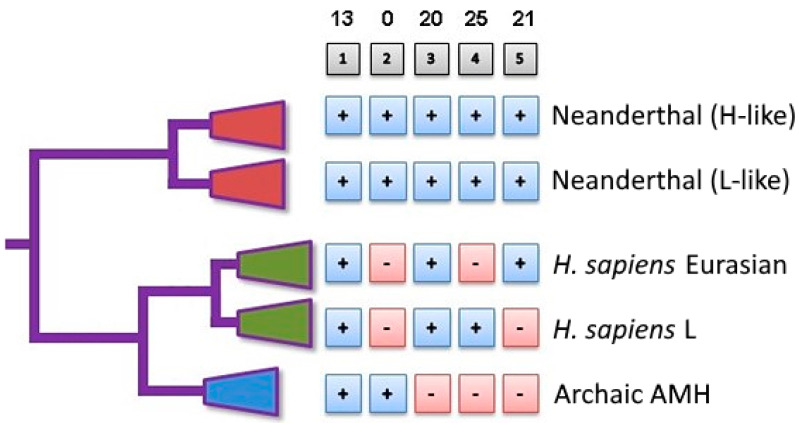
Phylogenetic distribution of Neanderthal mitogenome variants in present-day humans and archaic AMHs. The diagram depicts five patterns (gray boxes on top of columns) of N-SNVs in Neanderthals, archaic AMHs and the present-day *Homo sapiens* haplogroups L and Eurasian. The numbers above the gray boxes indicate the number of specific N-SNVs in the corresponding pattern. Plus (+) signs within blue boxes indicate the presence of N-SNVs in the corresponding clade and hyphen (-) in red boxes indicates absence of N-SNVs. Green clade indicates the Modern human lineage.

**Figure 5 ijms-25-03785-f005:**
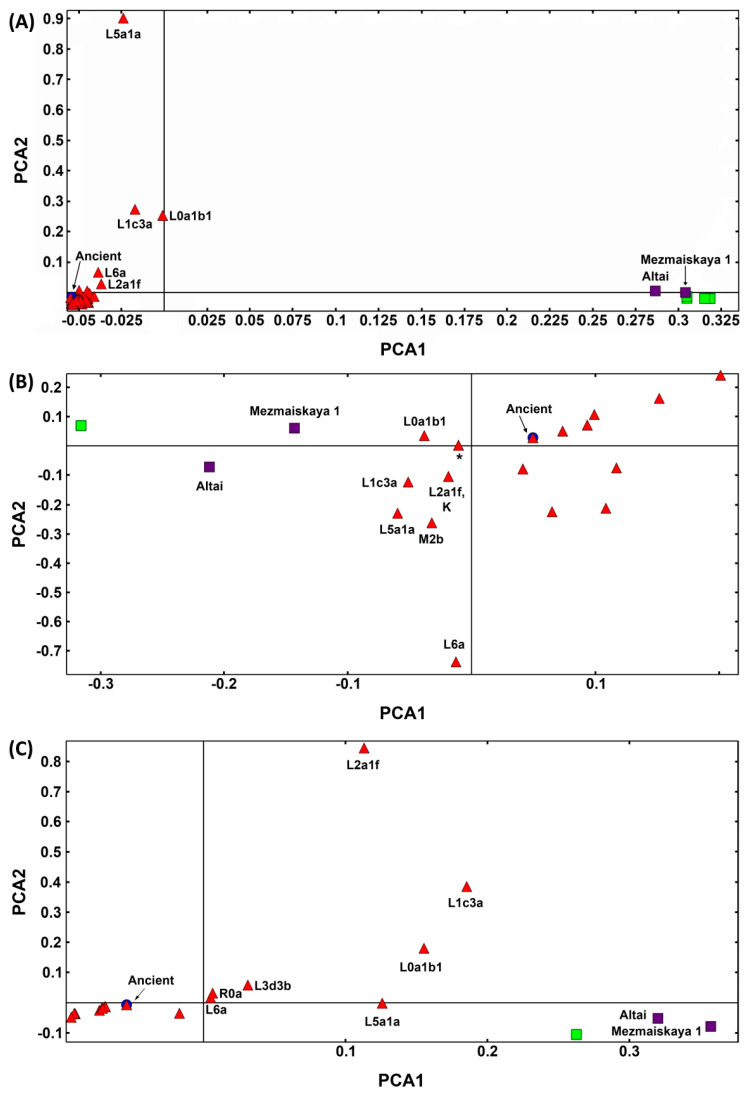
Principal component analysis (PCA) of human and Neanderthal mitogenomes. (**A**) PCA results for all 16,569 positions of 102 mitogenomes comprising 9 Neanderthals (green squares for H-like sequences and purple squares for L-like sequences), 41 archaic AMHs (blue circles) and 52 modern *H. sapiens* haplogroups (red triangles). The X-axis denotes the value for PC1, while the y-axis denotes values for PC2. Each dot in the figure represents one or more individuals. (**B**) PCA results for the segment comprising Positions 128 to 315 extracted from the 102 mitogenomes’ alignment, and (**C**) PCA results for the segment comprising Positions 6950 to 7660 extracted from the 102 mitogenomes’ alignment.

**Figure 6 ijms-25-03785-f006:**
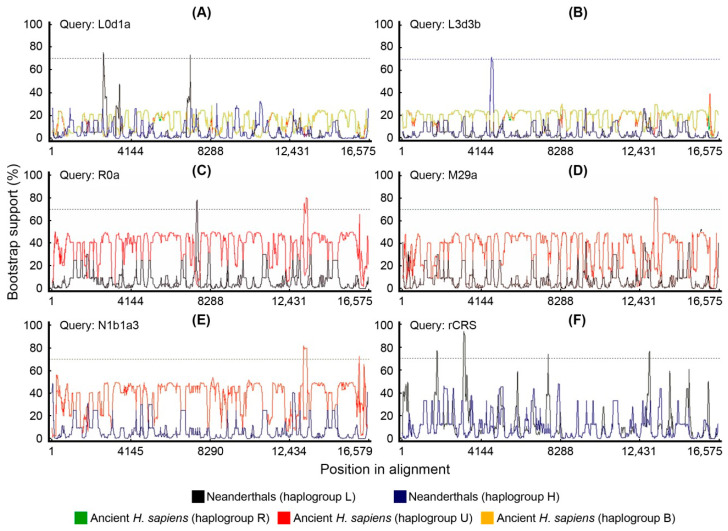
Bootscan recombination analysis. Possible recombination points were detected using different present-day mitogenome haplogroups as queries and ancient *H. sapiens* or Neanderthals as putative parental sequences. The same set of parental sequences were tested against query sequences of different haplogroups (**A**) L0d1a, (**B**) L3d3b, (**C**) R0a, (**D**) M29a, (**E**) N1b1a3 and (**F**) rCRS (H2a2a1).

**Figure 7 ijms-25-03785-f007:**
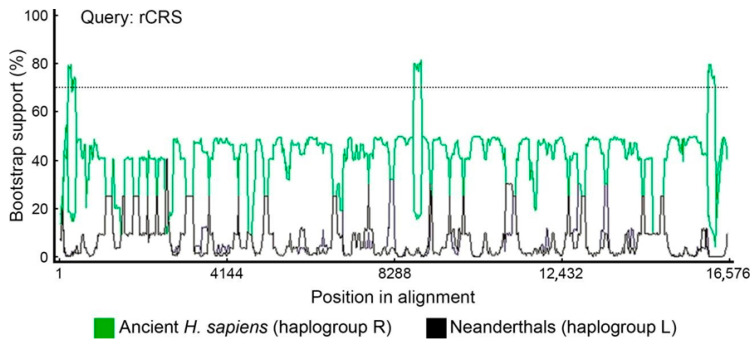
Bootscan analysis of the rCRS sequence (haplogroups H2a2a1) using ancient. *H. sapiens* and Neanderthals as putative parentals.

**Figure 8 ijms-25-03785-f008:**
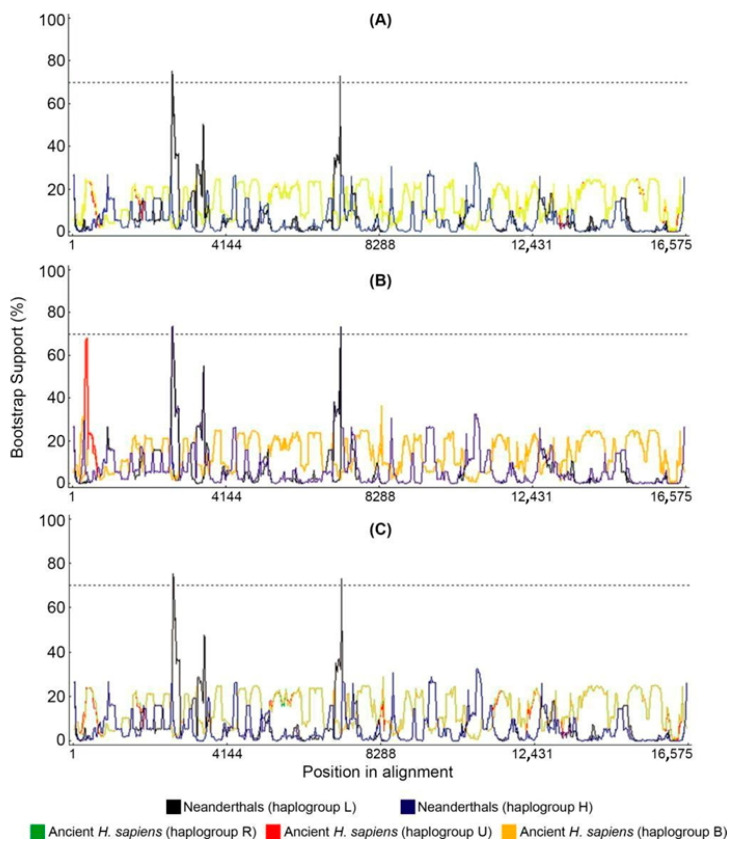
Bootscan/RDP analysis of haplogroup L0d1a showing the profile consistency regardless the alignment algorithm or bootscan model option. (**A**) Map to reference alignment and Felsenstein model on bootscan; (**B**) MAFFT alignment with Kimura two-parameters model on bootscan; (**C**) Map to reference alignment and Kimura two-parameters model on bootscan. Neanderthal sequences from the haplogroup L (black line) and haplogroup H (blue line) and archaic AMHs from the haplogroup R (green line), haplogroup U (red line) and haplogroup B (orange line) were used as parentals. The Neanderthal sequences were Feldhofer 1, Mezmaiskaya and Vindija33_16 and the archaic AMH sequences were Kostenki, Fumane, Doni Vestonice 14 and Tianyuan.

**Table 1 ijms-25-03785-t001:** Full list of all mitogenome N-SNVs analyzed in this study. Ts = transition, Tv = transversion, AA = amino acid, Nt = nucleotide, Freq. = frequency, Alig. = alignment, 1K Gen. = 1000 genomes project database, Prot. Effect = protein effect.

Position	DbSNP	Change	Change	Change	Type	Prot. Effect	Alignment.	1K Gen.	Gene	Codon
	rs	AA	Nt	Codon			Freq. (%)	Freq. (%)		Position
146	rs370482130	-	T>C	-	Ts	-	28.40%	21.40%	-	-
150	rs62581312	-	C>T	-	Ts	-	20.90%	19.10%	-	-
152	rs117135796	-	T>C	-	Ts	-	38.80%	33%	-	-
185	rs879015046	-	G>A	-	Ts	-	4.50%	3.70%	-	-
189	rs371543232	-	A>G	-	Ts	-	19.40%	7.70%	-	-
195	rs2857291	-	T>C	-	Ts	-	13.40%	24.70%	-	-
247	rs41334645	-	G>A	-	Ts	-	17.90%	7.00%	-	-
709	rs2853517	-	G>A	-	Ts	-	26.90%	16.70%	-	-
769	rs2853519	-	G>A	-	Ts	-	22.40%	16.40%	-	-
825	rs2853520	-	T>A	-	Tv	-	17.90%	7.10%	-	-
827	rs28358569	-	A>G	-	Ts	-	16.40%	4.30%	-	-
1018	rs2856982	-	G>A	-	Ts	-	22.40%	16.50%	-	-
2706	rs2854128	-	A>G	-	Ts	-	80.60%	89.00%	-	-
2758	rs2856980	-	G>A	-	Ts	-	6.00%	6.60%	-	-
2885	rs2854130	-	T>C	-	Ts	-	16.40%	6.80%	-	-
3010	rs3928306	-	G>A	-	Ts	-	17.90%	11.20%	-	-
3594	rs193303025	-	C>T	GTC>GTT	Ts	No change	20.90%	15.90%	*ND1*	3
4104	rs1117205	-	A>G	CTA>CTG	Ts	No change	19.40%	15.80%	*ND1*	3
4312	rs193303033	-	C>T	-	Ts	-	14.90%	1.80%	*tRNA*	-
4688	rs878853056	-	T>C	GCT>GCC	Ts	No change	4.50%	0.40%	*ND2*	3
5460	rs3021088	A>T	G>A	GCC>ACC	Ts	AA change	19.40%	7.50%	*ND2*	1
5471	rs879108598	-	G>A	ACG>ACA	Ts	No change	16.40%	0.50%	*ND2*	3
5821	rs56133209	-	G>A	-	Ts	-	14.90%	0.50%	*tRNA*	-
6962	rs1970771	-	G>A	CTG>CTA	Ts	No change	3.00%	3.00%	*COX1*	3
7146	rs372136420	T>A	A>G	ACT>GCT	Ts	AA change	6.00%	6.10%	*COX1*	1
7256	-	-	C>T	AAC>AAT	Ts	No change	20.90%	15.80%	*COX1*	3
7424	-	-	A>G	GAA>GAG	Ts	No change	16.40%	2.30%	*COX1*	3
7521	rs199474817	-	G>A	-	Ts	-	20.90%	16.00%	*tRNA*	-
7650	-	T>I	C>T	ACC>ATC	Ts	AA change	13.40%	0.00%	*COX2*	2
8386	-	-	C>T	ACC>ACT	Ts	No change	14.90%	0.00%	*ATP8*	3
8468	rs1116907	-	C>T	CTA>TTA	Ts	No change	16.40%	6.70%	*ATP8*	1
8655	-	-	C>T	ATC>ATT	Ts	No change	17.90%	7.10%	*ATP6*	3
9053	rs199646902	S>N	G>A	AGC>AAC	Ts	AA change	14.90%	2.60%	*ATP6*	2
9090	rs386829064	-	T>C	TCT>TCC	Ts	No change	3.00%	0.30%	*ATP6*	3
9755	rs2856985	-	G>A	GAG>GAA	Ts	No change	14.90%	2.00%	*COX3*	3
10,310	rs41467651	-	G>A	CTG>CTA	Ts	No change	16.40%	4.00%	*ND3*	3
10,373	rs28358277	-	G>A	GAG>GAA	Ts	No change	14.90%	3.00%	*ND3*	3
10,586	rs28358281	-	G>A	TCG>TCA	Ts	No change	3.00%	2.60%	*ND4L*	3
10,664	-	-	C>T	GTC>GTT	Ts	No change	14.90%	1.80%	*ND4L*	3
10,688	rs2853488	-	G>A	GTG>GTA	Ts	No change	17.90%	7.00%	*ND4L*	3
10,810	rs28358282	-	T>C	CTT>CTC	Ts	No change	17.9	7.10%	*ND4*	3
10,915	rs2857285	-	T>C	TGT>TGC	Ts	No change	14.90%	2.30%	*ND4*	3
11,914	rs2853496	-	G>A	ACG>ACA	Ts	No change	23.90%	12.70%	*ND4*	3
12,366	-	-	A>G	CTA>CTG	Ts	No change	14.90%	0.20%	*ND5*	3
12,810	rs28359174	-	A>G	TGA>TGG	Ts	No change	4.50%	2.40%	*ND5*	3
13,105	rs2853501	I>V	A>G	ATC>GTC	Ts	AA change	20.90%	13.70%	*ND5*	1
13,276	rs2853502	M>V	A>G	ATA>GTA	Ts	AA change	14.90%	1.80%	*ND5*	1
14,178	rs28357671	I>V	T>C	ATT>GTT	Ts	AA change	16.40%	5.20%	*ND6*	1
14,560	rs28357676	-	G>A	GTC>GTT	Ts	No change	16.40%	5.40%	*ND6*	3
15,148	rs527236206	-	G>A	CCG>CCA	Ts	No change	16.40%	1.10%	*CYTB*	3
15,244	rs28357369	-	A>G	GGA>GGG	Ts	No change	4.50%	2.10%	*CYTB*	3
15,355	rs527236181	-	G>A	ACG>ACA	Ts	No change	16.40%	0.60%	*CYTB*	3
16,086	rs386420030	-	T>C	-	Ts	-	4.50%	2.10%	-	-
16,093	rs2853511	-	T>C	-	Ts	-	13.40%	5.60%	-	-
16,148	rs201893071	-	C>T	-	Ts	-	17.90%	3.10%	-	-
16,169	rs879121566	-	C>T	-	Ts	-	1.50%	0.80%	-	-
16,182	rs879044186	-	A>C	-	Tv	-	15.20%	2.70%	-	-
16,183	rs28671493	-	A>G	-	Ts	-	1.50%	0.40%	-	-
16,209	rs386829278	-	T>C	-	Ts	-	17.90%	3.00%	-	-
16,230	rs2853514	-	A>G	-	Ts	-	14.90%	2.10%	-	-
16,234	rs368259300	-	T>C	-	Ts	-	16.40%	1.90%	-	-
16,278	rs41458645	-	C>T	-	Ts	-	25.40%	20.80%	-	-
16,298	rs148377232	-	T>C	-	Ts	-	11.90%	5.00%	-	-
16,311	rs34799580	-	T>C	-	Ts	-	34.30%	18.70%	-	-
16,320	rs62581338	-	C>T	-	Ts	-	16.40%	5.20%	-	-
16,519	rs3937033	-	T>C	-	Ts	-	61.20%	62.30%	-	-

**Table 2 ijms-25-03785-t002:** Disease-associated Neanderthal SNVs. Associations as compiled and summarized in MITOMAP [23].

Position	195	3010	5460	5821	16,093	16,183	16,519
Region/gene	D-loop	16S rRNA	*ND2*	tRNA-Cys	D-loop	D-loop	D-loop
dbSNP rs	rs2857291	rs3928306	rs3021088	rs56133209	rs2853511	rs28671493	rs3937033
DNA change	T>C	G>A	G>A	G>A	T>C	A>C	T>C
Type	Transition	Transition	Transition	Transition	Transition	Transversion	Transition
Codon position	-	-	1	-	-	-	-
Codon effect	-	-	Non-Syn.	-	-	-	-
Codon change	-	-	GCC>ACC	-	-	-	-
Protein change	-	-	Ala>Thr	-	-	-	-
Disease	Bipolar	Cyclic	Alzheimer’s	Deafness	Cyclic	Melanoma	Cyclic vomiting
associations	disorder/	vomiting	disease/	helper	vomiting	patients	syndrome with
	melanoma	syndrome with	Parkinson’s	mutation	syndrome		migraine/metastasis/
	patients	migraine	disease				glioblastoma, gastric,
							lung, ovarian, prostate
							tumors
Modern	L2a1f,	H1a1, J1c, D4	L0a1b1, L4a1,	C7	L4a1, R0a,	X1a, X3,	L1c3a, L2a1f, L6a,
haplogroups	L4a1,		W1, Q1		K1, H3, T1,	U1a1d, B2,	H15, H1a1, H3, I1, K,
	L5a1a, L6a,				A, C1a	M2b, M29a, O	K1, K2a2a, R0a1a3,
	W1, K,						R1a, U2c, V1a, W1,
	J2a2a, M2b						X1a, X3, B2, C4, C7,
							E1, F1a1, G1a1, M20,
							M29a, M2b, M3b,
							N1b1a3, N2a, O, P2
Neanderthal	Altai	Altai	Altai	Altai	Feldhofer1	El Sidron 1253	Altai
		El Sidron 1253	El Sidron 1253	El Sidron 1253	Vin. 33.25	Feldhofer 1	El Sidron 1253
		Feldhofer 1	Feldhofer 1	Feldhofer 1		Feldhofer 2	Feldhofer 1
		Feldhofer 2	Feldhofer 2	Feldhofer 2		Mezmaiskaya1	Feldhofer 2
		Mezmaiskaya1	Mezmaiskaya1	Mezmaiskaya1		Vindija 33.16	Mezmaiskaya1
		Vindija 33.16	Vindija 33.16	Vindija 33.16		Vindija 33.17	Vindija 33.16
		Vindija 33.17	Vindija 33.17	Vindija 33.17		Vindija 33.19	Vindija 33.17
		Vindija 33.19	Vindija 33.19	Vindija 33.19		Vindija 33.25	Vindija 33.19
		Vindija 33.25	Vindija 33.25	Vindija 33.25			Vindija 33.25

**Table 3 ijms-25-03785-t003:** Archaic AMH and Neanderthal mitogenomes analyzed in this study. Haplogroup inference quality (**) as calculated by Haplogrep 2 [25]. ENA * = European Nucleotide Archive (https://www.ebi.ac.uk/ena/browser/home).

	Archaic AMH	Haplogroup	Quality ** (%)	Age (Years)	Location	Reference	GenBank/ENA *
1	Berry Au Bac 1	U5b1a	98.78	7160–7319	France	[26]	KU534977
2	Bockstein	U5b1d1	98.72	8016–8329	Germany	[26]	KU534973
3	Cuiry Les Chaudardes 1	U5b1b	97.01	8050–8360	France	[26]	KU534975
4	Ofnet	U5b1d1	98.72	8159–8424	Germany	[26]	KU534974
5	Felsdach	U5a2c	97.43	8380–8980	Germany	[26]	KU534954
6	Hohlenstein Stadel	U5b2c1	94.77	8446–8809	Germany	[26]	KU534979
7	Falkenstein	U5b2a	92.19	8993–9409	Germany	[26]	KU534980
8	Mareuil Les Meaux 1	U5a2 + 16362	100	9080–9500	France	[26]	KU534959
9	Les Closeaux 3	U5a2	98.27	9580–10,230	France	[26]	KU534958
10	Ranchot 88	U5b1	95.95	9933–10,235	France	[26]	KU534978
11	Iboussieres 31-2	U5b1 + 16189	99.01	10,140	France	[26]	KU534976
12	Iboussieres 25-1	U5b2a	93.92	10,140	France	[26]	KU534981
13	Iboussieres 39	U5b2b	92.40	11,600–12,040	France	[26]	KU534972
14	Hohle Fels 10	U8a	96.92	12,700	Germany	[26]	KU534961
15	Rochedane	U5b2b	97.01	12,830–13,090	France	[26]	KU534971
16	Burkhardtshohle	U8a	96.92	14,150–15,080	Germany	[26]	KU534960
17	Hohle Fels 79	U8a	96.92	14,270–15,070	Germany	[26]	KU534962
18	Brillenhohle	U8a	97.66	14,400–15,120	Germany	[26]	KU534947
19	Oberkassel 998	U5b1+	99.45	14,000	Germany	[27]	KC521457
20	Goyet Q-2	U8a	96.92	14,780–15,230	Belgium	[26]	KU534963
21	Rigney 1	U2′3′4′7′8′9	87.68	15,240–15,690	France	[26]	KU534957
22	Hohle Fels 49	U8a	96.92	15,568–16,250	Germany	[26]	KU534964
23	Paglicci 71	U5b2b	97.97	18,197–18,973	Italy	[26]	KU534950
24	Dolni Vestonice 43	U5	97.44	25,000	Czech Rep.	[26]	KU534970
25	Goyet Q56-16	U2	91.40	26,040–26,600	Belgium	[26]	KU534965
26	Goyet 2878-21	U5	92.05	26,269–27,055	Belgium	[26]	KU534955
27	Goyet Q376-19	U2	89.43	27,310–27,720	Belgium	[26]	KU534967
28	Goyet Q55-2	U2	85.86	27,310–27,730	Belgium	[26]	KU534948
29	La Rochette	M	92.38	27,400–27,784	France	[26]	KU534951
30	Goyet Q53-1	U2	93.23	27,720–28,230	Belgium	[26]	KU534966
31	Paglicci 108	U2′3′4′7′8′9	92.40	27,831–28,961	Italy	[26]	KU534968
32	Paglicci 133	U8c	99.41	28,000–29,000	Italy	[26]	KU534956
33	Dolni Vestonice 16	U5	97.94	29,386–30,567	Czech Rep.	[26]	KU534949
34	Doni Vestonice 14	U5	100	31,000	Czech Rep.	[28]	KC521458
35	Cioclovina 1	U	96.72	32,519–33,905	Romania	[26]	KU534969
36	Goyet Q376-3	M	88.67	33,140–33,940	Belgium	[26]	KU534953
37	Goyet Q116-1	M	93.64	34,430–35,160	Belgium	[26]	KU534952
38	Kostenki 14	U2	92.69	36–39,000	Russia	[27]	FN600416 *
39	Fumane 2	R	94.43	39–41,000	Italy	[27]	KP718913
40	Tianyuan	B4′5	91.17	40,000	China	[29]	KC417443
41	Ust-Ishim	R	87.89	45,000	West Siberia	[27]	PRJEB6622 *
	**Neanderthals**						
1	Vindija 33.16	H1e	52.74	38,000	Croatia	[30]	AM948965
2	Vindija 33.17	H1e	52.75	Not dated	Croatia	[31]	KJ533544
3	Vindija 33.19	H1e	52.74	Not dated	Croatia	[31]	KJ533545
4	Vindija 33.25	H1as	52.84	Not dated	Croatia	[32]	FM865410
5	El Sidron 1253	H1e	52.81	39,000	Spain	[32]	FM865409
6	Feldhofer 1	H1as	52.84	40,000	Germany	[32]	FM865407
7	Feldhofer 2	H1e	52.81	40,000	Germany	[32]	FM865408
8	Altai Neanderthal	L1′2′3′4′5′6	55.96	50,000	Siberia	[3]	KC879692
9	Mezmaskaya 1	L1′2′3′4′5′6	52.12	65,000	Russia	[32]	FM865411

## Data Availability

All the data and files of analyses presented here are available upon request to the corresponding author.

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
