# Peer review of "Convergent Mutations and Single Nucleotide Variants in Mitochondrial Genomes of Modern Humans and Neanderthals"

_ijms, 2024, doi:10.3390/ijms25073785_

Round 1

Reviewer 1 Report

Comments and Suggestions for Authors

This manuscript from Ferreira and colleagues reports that the presence of N-SNVs in present-day human mitogenomes is due to homoplasy rather than recombination. The authors performed a thorough analysis to reach their overarching conclusion.

However, the manuscript must be improved prior to publication:

1) Throughout the manuscript, including the abstract, grammar,  syntax, and punctuation must be improved.

2) The tested hypotheses must be clearly enunciated at the end of the introduction.

3) A conclusive sentence is lacking at the end of each subsection of the Results section to inform readers, independent of their background in the subject.

4) The Discussion section needs to include an overall conclusive remarks and perspective to wrap up this study and to broadly describe the direction  of the future studies.

Comments on the Quality of English Language

The manuscript requires extensive editing. It is highly recommended that the authors retain an editing service.

Author Response

Reply to reviewer 1

This manuscript from Ferreira and colleagues reports that the presence of N-SNVs in present-day human mitogenomes is due to homoplasy rather than recombination. The authors performed a thorough analysis to reach their overarching conclusion.

R: We thank the reviewer for comments and suggestions

 However, the manuscript must be improved prior to publication:

1) Throughout the manuscript, including the abstract, grammar,  syntax, and punctuation must be improved.

R: The text was substantially revised. After correcting for structural problems (such as legend of Figure 4) the text was edited by Dr. Broach, coauthor who is native English speaker. The manuscript has tracked changes so the reviewer can readily verify the changed sections.

 2) The tested hypotheses must be clearly enunciated at the end of the introduction.

R: Corrected.

3) A conclusive sentence is lacking at the end of each subsection of the Results section to inform readers, independent of their background in the subject.

R: Corrected.

4) The Discussion section needs to include an overall conclusive remarks and perspective to wrap up this study and to broadly describe the direction  of the future studies.

R: Conclusive remarks and perspectives were included.

Reviewer 2 Report

Comments and Suggestions for Authors

Overall, this paper was strong. I was still a bit confused by Mueller ratchet but the rest of the conclusions and figures were acceptable. Except for figure 4, this is a bit confusing and I don't think it's necessary in this publication. I think this is a fun paper and I don't see any major glaring issues.

Author Response

Reply to reviewer 2

Overall, this paper was strong. I was still a bit confused by Mueller ratchet but the rest of the conclusions and figures were acceptable. Except for figure 4, this is a bit confusing and I don't think it's necessary in this publication. I think this is a fun paper and I don't see any major glaring issues.

R: The Muller's ratchet is a well established concept in population genetics and very well debated in the context of human mtDNA. References in the text discuss the problem extensively. For an excellent discussion please see: https://doi.org/10.1017/S0016672306008123

The manuscript has been substantially edited for formal structure and English.

Reviewer 3 Report

Comments and Suggestions for Authors

I would like to thank the Authors for the wonderful work done in this Manuscript, and the Editors for the opportunity to comment and have early glimpse to this wonderful new interpretation of Neanderthal "introgression" in modern mitochondrial DNA. Taking into consideration that, given the small size of the genome, it is much easier to accumulate mutations along the evolutionary timeline, I find that the authors have provided a poignant analysis of similarities in mtDNA between modern humans, "anatomically modern" ancient samples and the archaic ones. If anything, I think the introduction may be broadened a little by insisting on the topic of archaic genome introgression in modern humans, maybe providing a couple of the best known examples. I must say that Figure 3 may be a bit chaotic, and maybe some Figures and Tables (especially the bigger ones) may be put as Supplementary Material. Furthermore, Table numbering is incorrect, as Table 1 is indicated at page 17 of the manuscript.

Author Response

Reply to reviewr 3

I would like to thank the Authors for the wonderful work done in this Manuscript, and the Editors for the opportunity to comment and have early glimpse to this wonderful new interpretation of Neanderthal "introgression" in modern mitochondrial DNA. Taking into consideration that, given the small size of the genome, it is much easier to accumulate mutations along the evolutionary timeline, I find that the authors have provided a poignant analysis of similarities in mtDNA between modern humans, "anatomically modern" ancient samples and the archaic ones. If anything, I think the introduction may be broadened a little by insisting on the topic of archaic genome introgression in modern humans, maybe providing a couple of the best known examples. I must say that Figure 3 may be a bit chaotic, and maybe some Figures and Tables (especially the bigger ones) may be put as Supplementary Material. Furthermore, Table numbering is incorrect, as Table 1 is indicated at page 17 of the manuscript.

R: The manuscript has been edited for clarity and English.  Figures and legends have been improved and numbering of tables corrected. Supplementary tables were included.